# NoMaD: Goal Masked Diffusion Policies for Navigation and Exploration

**Ajay Sridhar, Dhruv Shah, Catherine Glossop, Sergey Levine**
UC Berkeley

**Abstract:** Robotic learning for navigation in unfamiliar environments needs to provide policies for both task-oriented navigation (i.e., reaching a goal that the robot has located), and task-agnostic exploration (i.e., searching for a goal in a novel setting). Typically, these roles are handled by separate models, for example by using subgoal proposals, planning, or separate navigation strategies. In this paper, we describe how we can train a single unified diffusion policy to handle both goal-directed navigation and goal-agnostic exploration, with the latter providing the ability to search novel environments, and the former providing the ability to reach a user-specified goal once it has been located. We show that this unified policy results in better overall performance when navigating to visually indicated goals in novel environments, as compared to approaches that use subgoal proposals from generative models, or prior methods based on latent variable models. We instantiate our method by using a large-scale Transformer-based policy trained on data from multiple ground robots, with a diffusion model decoder to flexibly handle both goal-conditioned and goal-agnostic navigation. Our experiments, conducted on a real-world mobile robot platform, show effective navigation in unseen environments in comparison with five alternative methods, and demonstrate significant improvements in performance and lower collision rates, despite utilizing smaller models than state-of-the-art approaches.

## 1 Introduction

Robotic learning provides us with a powerful tool for acquiring multi-task policies that, when conditioned on a goal or another task specification, can perform a wide variety of different behaviors. Such policies are appealing not only because of their flexibility, but because they can leverage data from a variety of tasks and domains and, by sharing knowledge across these settings, acquire policies that are more performant and more generalizable. However, in practical settings, we might encounter situations where the robot doesn't know *which* task to perform because the environment is unfamiliar, the task requires exploration, or the direction provided by the user is incomplete. In this work, we study a particularly important instance of this problem in the domain of robotic navigation, where the user might specify a destination visually (i.e., via a picture), and the robot must locate this destination by *searching* its environment. In such settings, standard multi-task policies trained to perform the user-specified task are not enough by themselves: we also need some way for the robot to explore, potentially trying different tasks (e.g., different possible destinations for searching the environment), before figuring out how to perform the desired task (i.e., locating the object of interest). Prior works have often addressed this challenge by training a separate high-level policy or goal proposal system that generates suitable exploratory tasks, for example using high-level planning [1], hierarchical reinforcement learning [2], and generative models [3]. However, this introduces additional complexity and often necessitates task-specific mechanisms. Can we instead train a single highly expressive policy that can represent *both* task-specific and task-agnostic behavior, utilizing the task-agnostic behavior for exploration and switching to task-specific behavior as needed to solve the task?

In this paper, we present a design for such a policy by combining a Transformer backbone for encoding the high-dimensional stream of visual observations with diffusion models for modeling a sequence of future actions and instantiate it for the particular problem of visual exploration and goal-seeking in novel environments. Our main insight is that such an architecture is uniquely suited for modeling task-specific and task-agnostic pathways since it provides high capacity (both for modeling perception and control) and the ability to represent complex, multimodal distributions.

The main contribution of our work is NoMaD, a novel architecture for robotic navigation in previously unseen environments that uses a unified diffusion policy to jointly represent exploratory

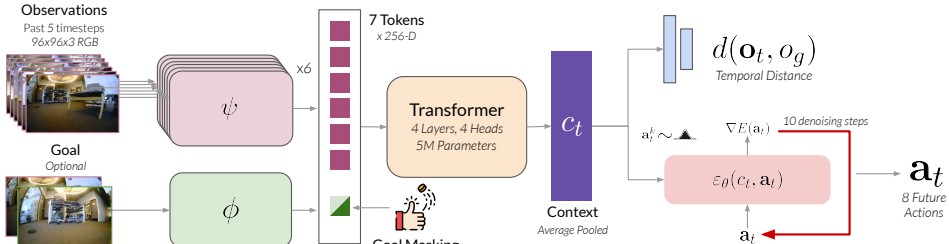

**Figure 1: Model Architecture.** NoMaD uses two EfficientNet encoders $\psi, \phi$ to generate input tokens to a Transformer decoder. We use *goal masking* to jointly reason about task-agnostic and task-oriented behaviors through the observation context $c_t$. We use action diffusion conditioned on the context $c_t$ to obtain a highly expressive policy that can be used in both a goal-conditioned and undirected manner.

task-agnostic behavior and goal-directed task-specific behavior in a framework that combines graph search, frontier exploration, and highly expressive policies. We evaluate the performance of NoMaD on both undirected and goal-conditioned navigation experiments across challenging indoor and outdoor environments, and report improvements over the state-of-the-art, while also being $15\times$ more computationally efficient. To the best of our knowledge, NoMaD is the first successful instantiation of a goal-conditioned action diffusion model, as well as a unified model for both task-agnostic and task-oriented behavior, deployed on a physical robot.

## 2 Preliminaries

Our objective is to design a control policy $\pi$ for visual navigation that takes the robot's current and past RGB observations as input $\mathbf{o}_t := o_{t-P:t}$ and outputs a distribution over future actions $\mathbf{a}_t := a_{t:t+H}$. The policy may additionally have access to an RGB image of a goal $o_g$, which can be used to specify the navigation *task*. When a goal $o_g$ is provided, $\pi$ must take actions that make progress towards the goal, and eventually reach it. In an unseen environment, the goal image $o_g$ may not be available, and $\pi$ must *explore* the environment by taking safe and reasonable navigation actions (e.g., avoiding obstacles, following hallways etc.), while providing sufficient coverage of the valid behaviors in the environment. To facilitate long-horizon exploration and goal-seeking, we follow the setup of ViKiNG [4] and pair $\pi(\mathbf{o}_t)$ with a topological memory of the environment $\mathcal{M}$, and a high-level planner that encourages the robot to explore the environment by navigating to unexplored regions. See Appendix B for more preliminaries.

## 3 Method

Unlike prior work that uses separate policies for goal-conditioned navigation and open-ended exploration [3], we posit that learning a single model for both behaviors is more efficient and generalizable. Training a shared policy across both behaviors allows the model to learn a more expressive prior over actions $\mathbf{a}_t$, which can be used for both conditional and unconditional inference. In this section, we describe our proposed NoMaD architecture, which is a goal-conditioned diffusion policy that can be used for both goal-reaching and undirected exploration. The NoMaD architecture has two key components: (i) attention-based goal-masking, which provides a flexible mechanism for conditioning the policy on (or masking out) an optional goal image $o_g$; and (ii) a diffusion policy, which provides an expressive prior over collision-free actions that the robot can take. Figure 1 shows an overview of the NoMaD architecture, and we describe each component in detail below.

### 3.1 Goal Masking

In order to train a shared policy for goal-reaching and undirected exploration, we modify the ViNT architecture described in Section 2 by introducing a binary "goal mask" $m$, such that $c_t = f(\psi(o_i), \phi(o_t, o_g), m)$. $m$ can be used to mask out the goal token $\phi(o_t, o_g)$, thus blocking the goal-conditioned pathway of the policy. We implement masked attention by setting the goal mask $m = 1$, such that the downstream computation of $c_t$ does not attend to the goal token. We implement unmasked attention by setting $m = 0$, such that the goal token is used alongside observation tokens in the downstream computation of $c_t$. During training, the goal mask $m$ is sampled from a Bernoulli

distribution with probability $p_m$. We use a fixed $p_m = 0.5$ during training, corresponding to equal number of training samples corresponding to goal-reaching and undirected exploration. At test time, we set $m$ corresponding to the desired behavior: $m = 1$ for undirected exploration, and $m = 0$ for reaching user-specified goal images. We find that this simple masking strategy is very effective for training a single policy for both goal-reaching and undirected exploration.

## 3.2 Diffusion Policy

While goal masking allows for a convenient way to condition the policy on a goal image, the distribution over actions that results from this, particularly when a goal is *not* provided, can be very complex. For example, at a junction, the policy might need to assign high probabilities to left and right turns, but low probability to any action that might result in a collision. Training a single policy to model such complex, multimodal distributions over action sequences is challenging. To effectively model such complex distributions, we use a diffusion model [5] to approximate the conditional distribution $p(\mathbf{a}_t|c_t)$, where $c_t$ is the observation context obtained after goal masking.

We sample a sequence of future actions $\mathbf{a}_t^K$ from a Gaussian distribution and perform $K$ iterations of *denoising* to produce a series of intermediate action sequences with decreasing levels of noise $\{\mathbf{a}_t^K, \mathbf{a}_t^{K-1}, \ldots, \mathbf{a}_t^0\}$, until the desired noise-free output $\mathbf{a}_t^0$ is formed. where $k$ is the number of denoising steps, $\epsilon_\theta$ is a noise prediction network parameterized by $\theta$, and $\alpha, \gamma$ and $\sigma$ are functions of the noise schedule.

The noise prediction network $\varepsilon_\theta$ is conditioned on the observation context $c_t$, which may or may not include goal information, as determined by the mask $m$. Note that we model the conditional (and not joint) action distribution, excluding $c_t$ from the output of the denoising process, which enables real-time control and end-to-end training of the diffusion process and visual encoder. During training, we train $\varepsilon_\theta$ by adding noise to ground truth action sequences. The predicted noise is compared to the actual noise through the mean squared error (MSE) loss.

## 4 Evaluation

We evaluate NoMaD in 6 distinct indoor and outdoor environments, and formulate our experiments to answer the following questions:

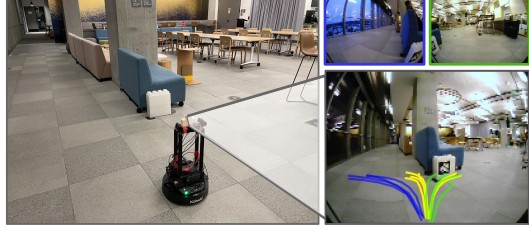

**Q1.** How does NoMaD compare to prior work for visual exploration and goal-reaching in real-world environments?

**Q2.** How does a joint task-agnostic and task-specific policy compare to the individual behavior policies?

**Figure 2:** Visualizing the task-agnostic (yellow) and goal-directed pathways for two goal images (green, blue) learned by NoMaD.

**Q3.** How important is the choice of visual encoder and goal masking to the performance of NoMaD?

## 4.1 Benchmarking Performance

*For more qualitative and quantitative results, see Appendix D.*

Towards understanding **Q1**, we compare NoMaD to six performant baselines for exploration and navigation in 6 challenging real-world environments. We follow the experimental setup of ViNT [3] and evaluate the methods on their ability to (i) explore a novel environment effectively in search of a goal position, or (ii) reach a goal indicated by an image in a previously explored environment, where the robot uses the policy to create a topological graph as episodic memory. All baselines are trained on a combination of GNM and SACSoN datasets for 20 epochs, and we perform minimal hyperparameter tuning to ensure stable training for each baseline. We report the mean success rate for each baseline, as well as the mean number of collisions per experiment.

Table 1 summarizes the results of our experiments in 5 challenging indoor and outdoor environments. VIB and Masked ViNT struggle in all the environments we tested and frequently end in collisions, likely due to challenges with effectively modeling multimodal action distributions. The Autoregressive

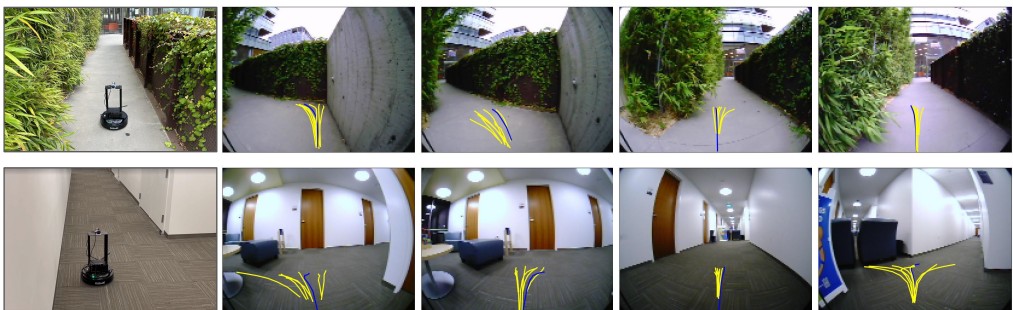

**Figure 3:** Visualizing rollouts of NoMaD deployed in challenging indoor (top) and outdoor (bottom) environments on the LoCoBot platform, showcasing successful exploration trajectories. Future action samples from the undirected mode are shown in yellow, and the action selected by the high-level planner is shown in blue. The predicted actions follow implicit navigational affordances, such as following hallways, and become multimodal at decision points, such as intersections in the hallway.

| Method | Params | Exploration | | Navigation |
| | | Success | Coll. | Success |
| --- | --- | --- | --- | --- |
| Masked ViNT[m] | 15M | 50% | 1.0 | 30% |
| VIB [6] | 6M | 30% | 4.0 | 15% |
| Autoregressive[m] | 19M | 90% | 2.0 | 60% |
| Random Subgoals [3] | 30M | 70% | 2.7 | **90%** |
| Subgoal Diffusion [3] | 335M | 77% | 1.7 | **90%** |
| NoMaD | 19M | **98%** | **0.2** | **90%** |

**Table 1:** NoMaD paired with a topological graph consistently outperforms all baselines for the task of exploration in previously unseen environments, and navigation in known environments. Most notably, NoMaD outperforms the state-of-the-art (Subgoal Diffusion) by 25%, while also avoiding collisions and requiring $15\times$ fewer parameters. [m]These baselines that use goal masking.

baseline uses a more expressive policy class and outperforms these baselines, but struggles in complex environments. Furthermore, the deployed policy tends to be jerky and slow to respond to dynamic obstacles in the environment, likely due to the discretized action space (see supplemental video for experiments). NoMaD consistently outperforms all baselines and results in smooth, reactive policies. For exploratory goal discovery, NoMaD outperforms the best published baseline (Subgoal Diffusion) by over 25% in terms of both efficiency and collision avoidance, and succeeds in all but the hardest environment. For navigation in known environments, using a topological graph, NoMaD matches the performance of the best published baseline, while also requiring a $15\times$ smaller model and running entirely on-the-edge. Figure 3 shows example rollouts of the NoMaD policy exploring unknown indoor and outdoor environments in search for the goal.

Analyzing the policy predictions across baselines (see Figure 4), we find that while the Autoregressive policy can (in principle) express multimodal distributions, the predictions are largely unimodal, equivalent to the policy learning the *average* action distribution. The Subgoal Diffusion baseline tends to represent the multiple modes well, but is not very robust. NoMaD consistently captures the multimodal distribution, and also makes accurate predictions when conditioned on a goal image.

## 5  Discussion

We presented NoMaD, the first instantiation of a goal-conditioned diffusion policy, that can perform both task-agnostic exploration and task-oriented navigation. Our unified navigation policy uses a high-capacity Transformer encoder with masked attention approach to flexibly condition on the task, such as goal images for navigation, and models the actions conditioned on observations using a diffusion model. We study the performance of this unified model in the context of long-horizon exploration and navigation in previously unseen indoor and outdoor environments, demonstrating over 25% improvements in performance over the state-of-the-art in previously unseen settings, while also requiring $15\times$ fewer computational resources.

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

## A  Related Work

Exploring a new environment is often framed as the problem of efficient mapping, posed in terms of information maximization to guide the robot to new regions. Some prior exploration methods use local strategies for generating control actions for the robots [7–10], while others use global strategies based on the frontier method [11–13]. However, building high-fidelity geometric maps can be hard without reliable depth information. Inspired by prior work [1, 14, 15], we factorize the exploration problem into (i) learned control policies that can take diverse, short-horizon actions, and (ii) a high-level planner based on a topological graph that uses the policy for long-horizon goal-seeking.

Several prior works have proposed learning-based approaches for robotic exploration by leveraging privileged information in simulation [16–19] or learn directly from real-world experience [6]. These policies have been trained with reinforcement learning to maximize coverage, predicting semantically rich regions, intrinsic rewards [16, 17, 20, 21], or by using planning in conjunction with latent variable and affordance models [6, 22, 23]. However, policies trained in simulation tend to transfer poorly to real-world environments [19, 24], and our experiments reveal that even the best exploration policies trained on real-world data perform poorly in complex indoor and outdoor environments.

The closest related work to NoMaD is ViNT, which uses a goal-conditioned navigation policy in conjunction with a *separate* high-capacity subgoal proposal model [3]. The subgoal proposal model is instantiated as a 300M parameter image diffusion model [5], generating candidate subgoal images conditioned on the robot's current observation. NoMaD uses diffusion models differently: rather than generating subgoal images with diffusion and conditioning on these generations, we directly model actions conditioned on the robot's observation with diffusion. Empirically, we find that NoMaD outperforms the ViNT system by over 25% in undirected exploration. Furthermore, since NoMaD does not generate high-dimensional images, it requires over $15\times$ fewer parameters, providing a more compact and efficient approach that can run directly on the less powerful onboard computers (e.g., NVIDIA Jetson Orin).

A key challenge with predicting sequences of robot actions for exploration is the difficulty in modeling multimodal action distributions. Prior work has addressed this by exploring different action representations, such as autoregressive predictions of quantized actions [25–28], using latent variable models [6, 23], switching to an implicit policy representation [29], and, most recently, using conditional diffusion models for planning and control [30–35]. State- or observation-conditioned diffusion models of action [31, 32] are particularly powerful, since they enable modeling complex action distributions without the cost and added complexity of inferring future states/observations. NoMaD extends this formulation by additionally conditioning the action distribution on both the robot's observations and the optional goal information, giving the first instantiation of a "diffusion policy" that can work in both goal-conditioned and undirected modes.

## B  Preliminaries

**Visual goal-conditioned policies:** To train goal-conditioned policies for visual inputs, we follow a large body of prior work on training high-capacity policies based on the Transformer architecture [3, 36–38]. Specifically, we use the Visual Navigation Transformer (ViNT) [3] policy as the backbone for processing the robot's visual observations $\mathbf{o}_t$ and goal $o_g$. ViNT uses an EfficientNet-B0 encoder [39] $\psi(o_i)$ to process each observation image $i \in \{t - P, \dots, t\}$ independently, as well as a goal fusion encoder $\phi(o_t, o_g)$, to obtain observation and goal tokens. These tokens are processed using a sequence of multi-headed attention layers $f(\psi(o_i), \phi(o_t, o_g))$ to obtain a sequence of context vectors that are concatenated to obtain the final context vector $c_t$. The context vector is then used to predict future actions $\mathbf{a}_t = f_a(c_t)$ and temporal distance between the observation and the goal $d(o_t, o_g) = f_d(c_t)$, where $f_a, f_c$ are fully-connected layers. The policy is trained using supervised learning using a maximum likelihood objective, corresponding to regression to the ground-truth actions and temporal distance. While ViNT shows state-of-the-art performance in goal-conditioned navigation, it cannot perform undirected exploration and requires another learned model for subgoal proposal. NoMaD extends ViNT to enable both goal-conditioned and undirected navigation.

**Exploration with topological maps:** While goal-conditioned policies can exhibit useful affordances and collision-avoidance behavior, they may be insufficient for navigation in large environments that require reasoning over long time horizons. To facilitate long-horizon exploration and goal-seeking in large environments, we follow the setup of ViKiNG [4] and integrate the policy with episodic memory $\mathcal{M}$ in the form of a topological graph of the robot's experience in the environment. $\mathcal{M}$ is represented by a graph structure with nodes corresponding to the robot's visual observations in the environment, and edges corresponding to navigable paths between two nodes, as determined by the policy's goal-conditioned distance predictions. When navigating large environments, the robot's visual observations $\mathbf{o}_t$ may not be sufficient to plan long-horizon trajectories to the goal. Instead, the robot can use the topological map $\mathcal{M}$ to plan a sequence of subgoals that guide the robot to the goal. When exploring previously unseen environments, we construct $\mathcal{M}$ online while the robot searches the environment for a goal. Beyond undirected coverage exploration, this graph-based

framework also supports the ability to reach high-level goals $G$, which may be arbitrarily far away and specified as GPS positions, locations on a map, language instructions, etc. In this work, we focus on frontier-based exploration, which tests the ability of NoMaD to propose diverse subgoals and search unseen environments. We largely follow the setup of prior work [4], swapping the learned policy with NoMaD.

## C    Training Details

The NoMaD model architecture is illustrated in Figure 1. We train NoMaD on a combination of GNM and SACSoN datasets, large heterogeneous datasets collected across a diverse set of environments and robotic platforms, including pedestrian-rich environments, spanning over 100 hours of real-world trajectories [40, 41]. NoMaD is trained end-to-end with supervised learning using the following loss function:

$$\mathcal{L}_{\text{NoMaD}}(\phi, \psi, f, \theta, f_d) = MSE(\varepsilon^k, \varepsilon_\theta(c_t, \mathbf{a}_t^0 + \varepsilon^k, k)) + \lambda \cdot MSE(d(\mathbf{o}_t, o_g), f_d(c_t)) \quad (1)$$

where $\psi, \phi$ correspond to visual encoders for the observation and goal images, $f$ corresponds to the Transformer layers, $\theta$ corresponds to the parameters of the diffusion process, and $f_d$ corresponds to the temporal distance predictor. $\lambda = 10^{-4}$ is a hyperparameter that controls the relative weight of the temporal distance loss. During training, we use a goal masking probability $p_m = 0.5$, corresponding to an equal number of goal-reaching and undirected exploration samples. The diffusion policy is trained with the Square Cosine Noise Scheduler [42] and $K = 10$ denoising steps. We uniformly sample a denoising iteration $k$, and we also sample a corresponding noise $\epsilon^k$ with the variance defined at iteration $k$. The noise prediction network, $\epsilon_\theta$, consists of a 1D conditional U-Net [30, 32] with 15 convolutional layers.

We use the AdamW optimizer [43] with a learning rate of $10^{-4}$ and train NoMaD for 30 epochs with a batch size of 256. We use cosine scheduling and warmup to stabilize the training process and follow other hyperparameters from ViNT [3]. For the ViNT observation encoder, we use EfficientNet-B0 [39] to tokenize observations and goals into 256-dimensional embeddings, followed by a Transformer decoder with 4 layers and 4 heads.

## D    More Results

### D.1    Baselines

**VIB:** We use the authors' implementation of a latent goal model for exploration [6], which uses a variational information bottleneck (VIB) to model a distribution of actions conditioned on observations.

**Masked ViNT:** We integrate our goal masking with the ViNT policy [3] to flexibly condition on the observation context $c_t$. This baseline predicts point estimates of future actions conditioned on $c_t$, rather than modeling the distribution.

**Autoregressive:** This baseline uses autoregressive predictions over a discretized action space to better represent multimodal action distributions. Our implementation uses a categorical representation of the action distribution, goal masking, and the same visual encoder design.

**Subgoal Diffusion:** We use the authors' implementation of the ViNT system [3] that pairs a goal-conditioned policy with an image diffusion model for generating candidate subgoal images, which are used by the policy to predict exploration actions. This is the best published baseline we compare against, but uses a $15\times$ larger model than NoMaD.

**Random Subgoals:** A variation of the above ViNT system which replaces subgoal diffusion with randomly sampling the training data for a candidate subgoal, which is passed to the goal-conditioned policy to predict exploration actions. This baseline does not use image diffusion, and has comparable parameter-count to NoMaD.

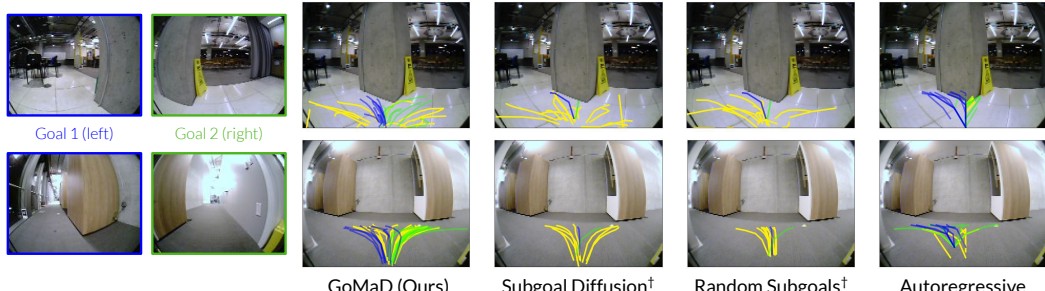

Goal 1 (left)    Goal 2 (right)                GoMaD (Ours)    Subgoal Diffusion†    Random Subgoals†    Autoregressive

**Figure 4:** Examples of action predictions from NoMaD and baselines in undirected mode (yellow) and goal-directed mode with two different goal images (blue towards left, green towards right). Only NoMaD can consistently represent multimodal undirected predictions while avoiding collisions with pillars or walls, as well as correctly predicting the goal-conditioned action predictions for the two goals. †Note that Subgoal Diffusion and Random Subgoals baselines only represent point estimates when conditioned on a goal image.

| Method | Params | Undirected | Goal-Conditioned |
|---|---|---|---|
| Diffusion Policy [32] | 15M | 98% | ✗ |
| ViNT Policy [3] | 16M | ✗ | 92% |
| NoMaD | 19M | 98% | 92% |

**Table 2:** Despite having comparable model capacities, NoMaD matches the performance of the best individual behavior policies for undireced exploration adn goal-conditioned navigation.

## D.2 Unified v/s Dedicated Policies

With the flexibility that a policy with task-specific and task-agnostic capabilities offers, **Q2** aims to understand the impact of goal masking on the individual behaviors learned by the policy. Specifically, we compare the performance of the jointly trained NoMaD model to the best-performing goal-conditioned and undirected models. We report the mean success rate for each baseline.

**Diffusion Policy:** We train a diffusion policy [32] with the same visual encoder as NoMaD and $m = 0$. This is the best exploration baseline, outperforming both VIB and IBC.

**ViNT Policy:** We use the authors' published checkpoint of the ViNT navigation policy [3], which predicts point estimates of future actions conditioned on observations and a goal. This is the best navigation baseline.

Comparing the unified NoMaD policy to the above, we find that despite having comparable model capacities, the unified policy trained with goal masking matches the performance of ViNT policy for goal-conditioned navigation and DP for undirected exploration. This suggests that training for these two behaviors involves learning shared representations and affordances, and a single policy can indeed excel at both task-agnostic and task-oriented behaviors simultaneously.

| Visual Encoder | Success | # Collisions |
|---|---|---|
| Late Fusion CNN | 52% | 3.2 |
| Early Fusion CNN | 68% | 1.5 |
| ViT | 32% | 2.5 |
| NoMaD | **98%** | **0.2** |

**Table 3:** The performance of our flexibly conditioned diffusion policy depends on the choice of visual encoder and goal masking strategy. The ViNT encoder with attention-based goal masking outperforms all alternatives.

### D.3 Visual Encoder and Goal Masking

We explore variations of the visual encoder and goal masking architectures to understand **Q3**. We consider two alternative visual encoder designs based on CNN and ViT backbones, and implement goal masking in different ways. We report the mean success rate for each baseline, as well as the mean number of collisions per experiment.

**Early/Late Fusion CNN:** We use convolutional encoders followed by an MLP to encode the observation and goal images, and use dropout on the goal embeddings followed by another MLP block to flexibly condition the observation context $c_t$ on the goal. $c_t$ obtained after dropout is used for conditioning the diffusion model in the same manner as NoMaD. We use a straight-through estimator [44] for propagating gradients to the observation and goal encoders during training. The goal can be combined with the observations either before or after the final MLP layers.

**ViT:** We divide the observation and goal images into $6 \times 6$ patches, and encode them using a Vision Transformer [45] into observation context $c_t$. For flexible conditioning, we use attention masks to block the goal patches from propagating information downstream.

We find the choice of visual encoder to be crucial for training diffusion policies, as summarized in Table 3. NoMaD outperforms both the ViT- and CNN-based architectures, successfully reaching the goal while avoiding collisions. CNN with early fusion outperforms late fusion, confirming similar analysis in prior work [3, 38], but struggles to effectively condition on goal information. Despite it's high capacity, the ViT encoder struggled learn a good policy, likely due to optimization challenges in training end-to-end with diffusion.

