# OpenReview forum: "NoMaD: Goal Masked Diffusion Policies for Navigation and Exploration"
_robot-learning.org/CoRL/2023/Workshop/OOD — OOD Workshop @ CoRL 2023_

### Official Review · Reviewer_Z4cA · 2023-10-16
**Combining task-specific and task-agnostic navigation policies**

**Rating:** 8
**Confidence:** 4

**Review:**

This paper provides a new architecture that combines task-agnostic policies for exploration and task-specific policies for goal attainment for navigation robots. The problem addressed involves navigating a robot to a goal that the robot has not seen before. The goal is communicated in the form of an image. The architecture combines the two policies with a common transformer backbone.

The paper is well written and the results are interesting. Navigating to new areas constitutes dealing with OOD scenarios, so the topic aligns with the theme of the workshop. I recommend accepting the paper.

---

### Official Review · Reviewer_YCgL · 2023-10-16

**Rating:** 7
**Confidence:** 3

**Review:**

### Summary:

This paper presents a network architecture for goal conditioned visuomotor navigation tasks, wherein a robot aims to navigate an environment to align it's visual input with a provided goal image. This task requires two phases, exploration, where the robot must navigate the environment (while avoiding collisions with walls, etc.) and store information that can be used to subsequently execute the goal-directed phase. In contrast to previous works which trained separate policies for exploration and goal directed navigation, the authors propose a transformer encoder which allows a single network to process inputs with an optional goal specification (which is masked in the case of exploration), together with a conditional diffusion model to learn potentially multimodal distributions over actions. The approach outperforms baselines both in exploring and navigating previously unseen environments.


### Strengths:
- The proposed architecture is novel wrt related work, and leverages the flexible distribution model capabilities of diffusion models to express both unconditional exploration policies and conditional navigation policies with a single network.
- The experimental results are promising, showing improvements relative to prior work. I also appreciated the comparisons against policies optimized specifically for exploration or goal-conditioned navigation, demonstrating that the network has the capacity to capture both distributions.

### Weaknesses:
- It was unclear what training objective was used, and what data was used for training. The authors state that the policies are trained via ground truth actions, but how is this ground-truth data collected? Requirements on expert goal-conditioned trajectories should be clearly stated.
- I would have liked to see more discussion of the generalization capabilities of the model. While evaluation environments were unseen, a more thorough evaluation of how the performance of this model generalizes (or doesn't) on environments or goal images that are OOD would strengthen the paper as well as its relevance to this workshop.
- Training the exploration policy directly via imitation learning could bias exploration towards the types of goals used by the experts in the ground truth trajectories. For example, in figure 2, the task-agnostic action distribution is biased towards going straight towards the pillar before turning, while the blue goal image requires turning sooner. The model seems to be able to generalize to this type of goal in this image -- exploring this generalization capability more quantitatively would provide useful insights.

---

### Decision · Program_Chairs · 2023-10-17

**Decision:**

Accept

**Comment:**

We agree with the reviewers’ assessment that this work is technically sound and will contribute to productive, topical discussions at the 2023 Workshop on OOD Generalization in Robotics. In particular, we appreciate that the authors consider a problem setting where OOD generalization is an important aspect of task completion, though we agree with the reviewers that generalization performance could be more specifically analyzed. We recommend the authors incorporate the reviewers’ feedback into their camera-ready submission to further improve their manuscript.